# A Swedish Genome-Wide Haplotype Association Analysis Identifies a Novel Breast Cancer Susceptibility Locus in 8p21.2 and Characterizes Three Loci on Chromosomes 10, 11 and 16

**DOI:** 10.3390/cancers14051206

**Published:** 2022-02-25

**Authors:** Elin Barnekow, Wen Liu, Hafdis T. Helgadottir, Kyriaki Michailidou, Joe Dennis, Patrick Bryant, Jessada Thutkawkorapin, Camilla Wendt, Kamila Czene, Per Hall, Sara Margolin, Annika Lindblom

**Affiliations:** 1Department of Clinical Science and Education, Södersjukhuset, Karolinska Institutet, 11883 Stockholm, Sweden; camilla.wendt@ki.se (C.W.); sara.margolin@ki.se (S.M.); 2Department of Oncology, Södersjukhuset, 11883 Stockholm, Sweden; per.hall@ki.se; 3Department of Molecular Medicine and Surgery, Karolinska Institutet, 17176 Stockholm, Sweden; wen.liu@neuro.uu.se (W.L.); hafdis.helgadottir@ki.se (H.T.H.); patrick.bryant@live.com (P.B.); jessada.thutkawkorapin@gmail.com (J.T.); 4Department of Neuroscience, Uppsala University, 75237 Uppsala, Sweden; 5Department of Clinical Genetics, Karolinska University Hospital, 17164 Stockholm, Sweden; 6The Cyprus Institute of Neurology & Genetics, Cyprus School of Molecular Medicine, 1683 Nicosia, Cyprus; kyriakimi@cing.ac.cy; 7Centre for Cancer Genetic Epidemiology, University of Cambridge, Cambridge CB18RN, UK; jgd29@cam.ac.uk; 8Department of Biochemistry and Biophysics, Stockholm University, 17165 Stockholm, Sweden; 9Science for Life Laboratory, 17165 Stockholm, Sweden; 10Department of Computer Engineering, Faculty of Engineering, Chulalongkorn University, Bangkok 10330, Thailand; 11Department of Medical Epidemiology and Biostatistics, Karolinska Institutet, 17165 Stockholm, Sweden; kamila.czene@ki.se

**Keywords:** *BNIP3L*, *FGFR2*, *TOX3*, *CASC16*, breast cancer, GWAS, haplotype

## Abstract

**Simple Summary:**

Heritable rare high- and moderate-risk mutations in breast cancer susceptibility genes are known of, alongside 170 common genetic low risk variants with a minor increase in risk. However, based on genetic studies, we know that over half of the breast cancer heritability is still unexplained. By analyzing combinations of chromosomal nearby variants, so-called haplotypes, and their association to breast cancer we could identify a novel genetic breast cancer risk locus on chromosome 8 and confirm three well known low risk loci on Chr 10, 11 and 16.

**Abstract:**

(1) Background: The heritability of breast cancer is partly explained but much of the genetic contribution remains to be identified. Haplotypes are often used as markers of ethnicity as they are preserved through generations. We have previously demonstrated that haplotype analysis, in addition to standard SNP association studies, could give novel and more detailed information on genetic cancer susceptibility. (2) Methods: In order to examine the association of a SNP or a haplotype to breast cancer risk, we performed a genome wide haplotype association study, using sliding window analysis of window sizes 1–25 and 50 SNPs, in 3200 Swedish breast cancer cases and 5021 controls. (3) Results: We identified a novel breast cancer susceptibility locus in 8p21.1 (OR 2.08; *p* 3.92 × 10^−8^), confirmed three known loci in 10q26.13, 11q13.3, 16q12.1-2 and further identified novel subloci within these three loci. Altogether 76 risk SNPs, 3302 risk haplotypes of window size 2–25 and 113 risk haplotypes of window size 50 at *p* < 5 × 10^−8^ on chromosomes 8, 10, 11 and 16 were identified. In the known loci haplotype analysis reached an OR of 1.48 in overall breast cancer and in familial cases OR 1.68. (4) Conclusions: Analyzing haplotypes, rather than single variants, could detect novel susceptibility loci even in small study populations but the method requires a fairly homogenous study population.

## 1. Introduction

The inherited genetic background of breast cancer is only partly understood. Epidemiological studies have shown breast cancer to be approximately twice as common in first degree relatives of breast cancer patients compared to the general population [1]. There are well known inherited genetic risk factors that contribute to breast cancer susceptibility. High to moderate penetrant susceptibility genes—BRCA1, BRCA2, P53, PALB2, ATM and CHEK2—explain approximately 30% of heritability in familial breast cancer cases and to date more than 170 low risk loci, associated with a minor increase in risk, have been identified accounting for another 18% [2,3,4,5,6,7,8]. Over half of the heritability is thus still unexplained.

The majority of known low risk loci have been identified through large Genome Wide Association Studies (GWAS), based on Single Nucleotide Polymorphism (SNP) association, conducted within the Breast Cancer Association Consortium (BCAC) [8,9,10]. SNPs that are located in close proximity to each other on a chromosome (Chr) tend to be inherited together and constitute a haplotype. Haplotypes are often used as ethnicity markers as they differ between populations. Therefore, homogenous populations are preferable to examine haplotype effects on a trait. Sweden has a fairly homogenous population and founder mutations in cancer genes are known [11]. Haplotype association has not been widely applied in cancer susceptibility studies. However, we have previously used this approach in identifying novel cancer susceptibility loci in the Swedish population [12,13,14,15]. In a GWAS of 3555 twins affected with cancer and 15,581 unrelated twins without cancer, seven susceptibility loci with increased general cancer risk were suggested [12]. We have also used haplotype analysis in familial cancer studies, where BCAC genotype data were analyzed to confirm and complete a candidate breast cancer region on chromosome 6q14.1 [13]. A similar study used genotype data from a collaboration of colorectal cancer consortium, CORECT, to perform haplotype analysis of a colorectal cancer risk locus on chromosome 9q22.32 [14]. Recently, haplotype analysis was also used in our search for a suggested novel cancer syndrome with increased risk of colorectal, gastric and prostate cancer [15]. Hence, we hypothesized that the Swedish population would be sufficiently homogeneous for haplotype analysis and that this approach could give additional information compared to SNP analysis in identifying breast cancer risk loci.

The aim of this study was to identify breast cancer risk haplotypes in the Swedish population. Therefore, we were prompted to reanalyze two Swedish breast cancer cohorts—KARMA and KARBAC—consisting of 3200 invasive breast cancer cases and 5021 controls previously analyzed within the BCAC consortium [8,9,10,16,17,18,19]. SNP and haplotype analysis was performed in the search for founder haplotypes and compared with existing SNP results from BCAC. As genetic risk is enriched in familial cases, subgroup analysis on familial cases was performed.

## 2. Materials and Methods

### 2.1. Study Population

We included 3215 invasive breast cancer cases from two Swedish cohorts, KARMA (*n* = 2712) and KARBAC (*n* = 503), and controls without family history of breast cancer from KARMA (*n* = 5032) in the analysis. The cohorts are thoroughly described elsewhere and previously analyzed in several BCAC studies [8,9,10,16,17,18,19,20,21,22]. Briefly, the KARMA Cohort consists of 70,877 women performing a screening or clinical mammogram at 4 hospitals in Sweden during the period October 2010–March 2013 [20]. The KARBAC cohort includes two sub-cohorts of breast cancer cases. KARBAC1 (*n* = 394) is a hospital cohort of consecutive breast cancer cases recruited from October 1998 until May 2000 [21]. KARBAC2 (*n* = 109) is a BRCA1- and BRCA2 negative breast cancer cohort recruited from a clinical genetic counselling department from February 2000 until January 2012 [22]. The studies were approved by the local ethical board and all individuals gave written informed consent.

For the subgroup analysis on familial breast cancer, we identified 652 self-reported familial cases in the merged KARMA and KARBAC cohort (516 familial KARMA breast cancer cases and 136 familial KARBAC breast cancer cases). Familial breast cancer cases were defined as individuals affected by breast cancer with a first degree relative diagnosed with breast cancer. The same KARMA controls, as in the overall breast cancer study, were used.

### 2.2. Genotyping and Quality Control 

DNA was extracted from peripheral blood samples using standard procedures. DNA from 8247 individuals were genotyped with an Illumina Infinum OncoArray-500K B BeadChip [8,23,24]. A total of 474,706 SNPs were shared between the cohorts. The datasets were merged using PLINK v.1.9 [25,26]. TOP strand format was accounted for. Quality Control (QC) was performed and variants were excluded if call rates <98% (2332 variants removed), minor allele frequency <0.01 (138,834 markers removed) or deviation from Hardy–Weinberg equilibrium *p* < 0.001 (634 markers removed). No individual was removed due to missing genotype data. In the last step of QC a Multi-Dimensional Scaling (MDS) analysis was conducted to identify ethnic outliers [27]. The MDS coordinates were also used in the logistic regression model (see Section 2.3) to adjust for population stratification. Individuals above level +0.04 and below −0.04 in the MDS analysis for the four dimensions—Coordinate1 (C1), C2, C3 or C4 are defined as ethnic outliers (13 sporadic, 2 familial breast cancer cases and 11 controls) and were excluded from the dataset. The remaining individuals were plotted in a MDS plot (Appendix A). A total of 332,906 variants, 3200 breast cancer cases (2550 sporadic; 650 familial) and 5021 controls passed QC and remained for further downstream analyses. Reference panel GRCH37 was applied for SNPs and chromosomal positions.

### 2.3. Statistics

We performed a sliding window haplotype GWAS using PLINK 1.07 from a window size of (wdw) 1 to 25 SNPs using a logistic regression model to examine the effect of a SNP or a haplotype of various length on the risk of breast cancer [26,28]. In the first analysis. overall breast cancer was set as outcome and in the second analysis familial breast cancer was used. The dataset from the first analysis was reanalyzed for haplotypes with a window size of 50 SNPs on Chr 10, 11 and 16. Adjustment for population stratification with principal coordinates C1, C2, C3 and C4 from MDS analysis (see QC procedure) as covariates in the logistic regression model was performed. For SNP analysis, genomic inflation factor was calculated (λ = 1.04). Population stratification was not considered a confounding factor. The effect of exposure on outcome was estimated by odds ratio (OR). Only OR >1 is reported in this study as the aim was to identify risk loci. The *p* value criteria for genome-wide statistical significance was applied, i.e., *p* < 5 × 10^−8^ was considered statistically significant [29]. No correction for multiple testing was performed as we assumed that all haplotypes of various length in each sublocus reflected the same genetic risk locus. A quantile–quantile (QQ) plot was created for single SNP association (Appendix A). A Manhattan plot was created to display observed *p*-values along the chromosomes (Appendix A). The QQ and Manhattan plots were generated in R using the qqman package. The haplotype positions were mapped and transposed using a custom script (https://github.com/patrickbryant1/CMM/blob/master/hap_vis.py) (accessed on 24 February 2022) after which the haplotypes were sorted on OR and illustrated in Figure 1, Figure 2 and Figure 3. Clusters of haplotypes with a certain variant in focus in Figure 1, Figure 2 and Figure 3 are called “subloci” in this study and considered to represent the same chromosomal risk locus.

## 3. Results

We identified four susceptibility loci on Chr 8, 10, 11 and 16 with a support of, in total, 76 risk SNPs (illustrated in a Manhattan plot, Appendix A), 3302 risk haplotypes of window size 2–25 and 113 risk haplotypes of window size 50 in a Swedish GWAS dataset of 3200 breast cancer cases and 5021 controls. Table 1 presents SNPs and haplotypes with the lowest *p*-value for each locus on Chr 8, 10, 11 and 16. All significant breast cancer risk SNPs and haplotypes of various lengths on Chr 10, 11 and 16 were sorted based on OR and are illustrated in Figure 1a, Figure 2a and Figure 3a. In each locus the SNP with the lowest *p*-value is indicated with a star, which chromosomal position is outlined in red, and the haplotype with the lowest *p*-value is indicated with a black rectangle. The position of previous published SNPs from the BCAC collaboration, in this study called “BCAC SNPs”, are outlined in yellow. In the plots, the x-axes are compressed for the loci including many haplotypes to be able to illustrate them all. All overlapping significant SNPs and haplotypes have identic alleles on a given position. The subloci (see Section 2) within the loci on Chr 10, 11 and 16 are described in more detail for each locus below. 

### 3.1. Locus 8p21.2

We identified one significant haplotype but no significant SNP (Table 1) in a novel breast cancer risk locus, 8p21.2 (Appendix A). In the subgroup analysis of familial cases the OR increased from 2.08 to 2.33 but did not reach significant level, *p* = 6. 97 × 10^−5^ (Appendix A). The significant haplotype is located within the *BNIP3L* gene. 

### 3.2. Locus 10q26.13

We identified 26 significant risk SNPs and 361 significant risk haplotypes in 10q26 (Appendix A). Their positional relation is visualized in Figure 1a. According to the haplotype clusters illustrated in Figure 1a, four separate risk subloci were suggested in locus 10q26.13: Sublocus 10q(a) is above the red line, around the significant haplotype with the lowest *p*-value (Table 1); sublocus 10q(b) is on the red line and the significant SNP with the lowest *p*-value, rs2912780 (Table 1); Sublocus 10q(c) is on the upper yellow line and the two nearby published BCAC SNPs, rs2981578 and rs35054928 (Table 2); sublocus 10q(d) is on the bottom yellow line and the published BCAC SNP rs45631563 (Table 2). The three SNPs included in the haplotype with the lowest *p*-value on Chr 10, all individually showed significance in single SNP analysis. However, when combined in the above haplotype, OR increased from 1.22 to 1.31 in single SNP analysis to 1.36 in haplotype analysis. In addition, the haplotype of window size 50 with the lowest *p*-value (Table 1), of which the first and last SNP are indicated by dashed lines in Figure 1a, included all four subloci and with a higher OR, 1.46 (non-significant). The subgroup analysis of familial cases showed similar pattern but with increased OR 1.54 compared to the analyses of all cases 1.36 (Figure 1b, Appendix A). All SNPs and haplotypes at this locus are located within the *FGFR2* gene.

### 3.3. Locus 11q13.3 

We identified one significant SNP and 500 risk haplotypes (481 wdw 2-25, 19 wdw 50) in 11q13.3 (Appendix A). Similarly, as for Chr 10, their positional relation are illustrated in Figure 2a. According to the haplotype clusters two separate risk subloci were suggested in locus 11q13.3: sublocus 11q(a) on the red line with the only significant risk SNP rs614367 (Table 1) and sublocus 11q(b) on the bottom yellow line and the published BCAC SNP rs75915166 (Table 2). The haplotype with the lowest *p*-value (Table 1) within this locus includes the significant SNP but none of the two published BCAC SNPs. The vast majority of significant haplotypes showed a higher OR than the significant SNP. The significant haplotype of window size 50 with the lowest *p*-value (Table 1), is indicated by dashed lines in Figure 2a and involve only the upper sublocus. The analysis of familial samples resulted in a few statistically significant haplotypes, all around rs614367 and no haplotypes including any of the published BCAC SNPs (Figure 2b) were identified. Again, the OR was increased in the familial subgroup analysis, (Appendix A), compared to the analysis of all breast cancer cases, with OR 1.68 and 1.48, respectively. The locus on Chr 11 is located upstream of the CCND1 gene.

### 3.4. Locus 16q12.1-16q12.2

We identified 49 significant risk SNPs and 2553 significant risk haplotypes (2459 wdw 2-25, 94 wdw 50) in 16q12.1-16q12.2 (Appendix A). Similarly, as for Chr 10 and 11, SNPs and haplotypes of various lengths and their positional relation are visualized in Figure 3a. According to the haplotype clusters illustrated in Figure 3a five separate risk subloci are suggested in locus 16q12.1-16q12.2: sublocus 16q(a) is above the red line; sublocus 16q(b) is on the red line with the significant SNP rs12918816 with the lowest *p*-value (Table 1); sublocus 16q(c) is between the red and yellow line; sublocus 16q(d) is on the yellow line with the published BCAC SNP rs4784227, (Table 2) and sublocus 16q(e) is below the yellow line in Figure 3a. The risk haplotype with the lowest *p*-value is a haplotype consisting of two SNPs including the significant SNP with the lowest *p*-value but not the BCAC SNP. The haplotype with the highest OR does not include the BCAC SNP. However, the haplotype of window size 50 with the lowest *p*-value (Table 1) includes both the significant Swedish SNP with the lowest *p*-value and the published BCAC SNP. In the familial subgroup analysis, the result showed a similar pattern (Figure 3b). Again, the OR was increased in the familial subgroup analysis, OR 1.49 (Appendix A), compared to the analysis of all breast cancer cases, OR 1.39. Subloci 16q(a–c) are located within the *TOX3* gene while Subloci 16(d,e) within the *CASC16* gene.

### 3.5. Validation of Material and Methods Used

In order to increase power only invasive cases and controls without a family history of breast cancer were used in our study. To test the hypothesis that removing cancer in situ and familial controls would increase power the full data set, including cancer in situ and familial controls, was reanalyzed for Chr 8, 10, 11 and 16 with similar results for Chr 10, 11 and 16, whereas the locus on Chr 8 was not identified.

## 4. Discussion

In the search for founder haplotypes, we reanalyzed genotype data from Swedish samples included in the BCAC by performing both a haplotype and SNP GWAS of 3200 Swedish invasive breast cancer cases and 5021 controls. We identified a novel locus on Chr 8 and confirmed and fine-mapped three of the known breast cancer risk loci on Chr 10, 11 and 16 with haplotype analysis. Within the three known loci, novel subloci were identified. Top risk haplotypes presented higher ORs than top risk SNPs at the same locus. The novel breast cancer susceptibility locus 8p21.2 has been discussed before in a cancer perspective. Zeegers et al. presented strong evidence for 8p21.1-8p21.3 to harbor prostate cancer susceptibility genes, which confirmed previous prostate cancer GWAS findings [33]. The gene *BNIP3L* is located in the region of the novel locus and encodes a proapoptotic protein [34]. Loss of heterozygosity (LOH) is demonstrated in 20% of sporadic breast cancer, and in 40% in familial breast cancer in 8p12-22 [35,36]. Lai et al. used gene expression analysis and direct sequencing in 25 breast cancer cases and could not find support for *BNIP3L* as a breast cancer gene, why nearby genes or transcripts are suggested to be the target for LOH at 8p21 in breast cancer [37]. However, the *BNIP3L* gene is still of interest and a recent publication suggested a *BNIP3L* dependent pathway to be a target for cancer therapy in a subgroup of triple negative breast cancer [38]. The genes involved in the known three loci—*FGFR2*, *CCND1* and *TOX3*—have been thoroughly discussed in relation to breast cancer [32,39,40,41,42]. Fine mapping of the loci on Chr 10, 11 and 16 based on BCAC data by Fachal et al. suggest causal variants close to previously published BCAC SNPs in the three regions [19]. Interestingly the BCAC SNP—rs4784227 is located outside *TOX3*, in the area of the *CASC16* gene, but considered to be causative through a *TOX3* gene effect, whereas three of our novel subloci on Chr 16 are located within the *TOX3* gene itself [32]. In contrast to haplotype analysis, where the pathogenic variants are suggested to lie within the borders of the risk haplotypes, SNP analysis is not able to indicate the distance between the risk SNP and the pathogenic variant(-s). Different approaches are therefore being addressed to improve the “nearest gene” approach in SNP analysis. A recent publication studied associations between SNPs, levels of gene expression (eQTLs), disease-specific survival and somatically mutated cancer genes to define putative target loci, including 10q26.13, 11q13.3 and 16q12.1-16q12.2 [43]. The results suggested the same genes as in our study, *FGFR2* on 10q26.13 and *CCND1* on 11q13.3, while the relation to the gene *TOX3* on 16q12.1-16q12.2 was less clear. This could possibly relate to our finding of two loci at this target, *TOX3* and *CASC16*. The genes *FGFR2*, *CCND1* and *TOX3* are still not studied much in relation to the roles in breast cancer although genetic studies have suggested their involvement in cancer risk. *FGFR2* has been suggested to be implicated in breast cancer as a target for therapeutic strategies [44], *CCND1* is frequently mutated by copy number variation and suggested as a prognostic biomarker [45]. The role of *TOX3* in relation to breast cancer is less clear. However, *TOX3* has been shown to stimulate estrogen in ovarian granulosa cells, suggesting a role in breast cancer pathogenesis [46]. Moreover, a recent study published a variant in *CASC16* to correlate to breast cancer susceptibility [47].

Haplotype analysis could identify a novel breast cancer susceptibility locus in 8p21.2, whereas SNP analysis could not. The haplotype in 8p21.2 is rarer than the haplotypes in the known loci and could be a founder haplotype that happens to have reached a frequency that makes it detectable. When unfiltered cases and controls were used for analysis, the region was not identified, which could be explained by the increased power of the filtered analysis where only invasive cases and controls without a family history of breast cancer were used. Replication of this locus is required for confirmation, but previous findings in other tumor types in the Swedish population support the use of haplotype analysis, in addition to SNP analysis [12,13,14,15]. Furthermore, to find the actual risk causing variant, sequencing of familial breast cancer patients with this particular haplotype would be ideal. A haplotype is expected to better identify a founder risk allele than an SNP. Generally, higher ORs were presented for haplotypes than for SNPs at a certain locus. Haplotype analysis could therefore be preferable in small populations in identifying novel susceptibility loci. Moreover, wider haplotypes generally showed higher OR rather than smaller, supporting the idea of more than one risk variant being involved at one locus. However, the OR for the significant haplotype of window 50 with the lowest *p*-value on Chr 16 was similar to the OR of the significant single SNP with the lowest *p*-value. The haplotype effect is suggested to depend on whether nearby SNPs act together in a risk generating or protective mode. You could therefore hypothesize that protective subloci could be involved as well at the locus on Chr 16. The risk SNPs were consistent with previous BCAC results (Appendix A). Enrichment of genetic risk alleles are expected in familial cases. Familial analysis confirmed a genetic association by showing higher OR in the three known loci. Single SNP analysis showed somewhat higher ORs compared to BCAC, which is illustrated in Table 2 (for comparison all published BCAC SNPs see Appendix A). This could be explained by including only invasive cases and controls with no family history of breast cancer in our study.

This study has a smaller sample size compared to previous published breast cancer GWASs within the BCAC collaboration including our samples [8,9,10,16,17,18,19]. However, the aim of this study was to demonstrate an alternative approach in finding risk loci—haplotype GWAS in a fairly homogeneous population, in comparison to previous single SNP GWAS with merged samples from several populations within the BCAC collaboration. We identified one novel locus on Chr 8, and several novel subloci within three well-known risk loci with immense support (3490 statistically significant risk SNPs and -haplotypes). However, we could only confirm three loci, including 6 published SNPs (Table 2), out of more than 170 susceptible SNPs previously identified (Suppl. T6). This low replication rate could be explained by the relatively small sample size, underpowered to detect very low risk loci, even if power has been suggested to be increased when using haplotype—rather than SNP analysis [12]. In addition, clinical relevance of these very low risk loci can be discussed. Even if there is clear support for founder variants within the known loci on Chr 10, 11 and 16, we present diversity between the Swedish and overall BCAC populations regarding loci on Chr 10, 11 and 16, which is illustrated in Figure 1a, Figure 2a and Figure 3a. Global diversity of cancer susceptibility loci is known [48]. We suggest, as the subloci are fairly separated, that they probably represent diverse mutations, and further studies are warranted.

The level chosen for statistical significance could be discussed. Haplotype analysis include multiple testing, but numerous overlapping haplotypes are generated. We assume that all haplotypes in each sublocus reflect the same genetic risk locus, and therefore no correction for multiple testing was performed.

## 5. Conclusions

In conclusion, this haplotype GWAS, using Swedish breast cancer cases and controls, demonstrated that haplotype analysis could give novel and more detailed information on risk variants in addition to single SNP analysis. However, haplotype analysis most often requires a fairly homogenous population. Genetic risk is enriched in familial cases and thus haplotype analysis in homogenous populations of familial cases could potentially identify novel risk loci even in small study populations. We identified a novel breast cancer risk locus in 8p21. Further replication of this locus is required. Most importantly, there is clear support of founder variants at the three known loci on Chr 10, 11 and 16, but our study did not support any other previously reported loci. Future studies in haplotype regions could define causative variants to be used in genetic testing and risk prediction.

## Figures and Tables

**Figure 1 cancers-14-01206-f001:**
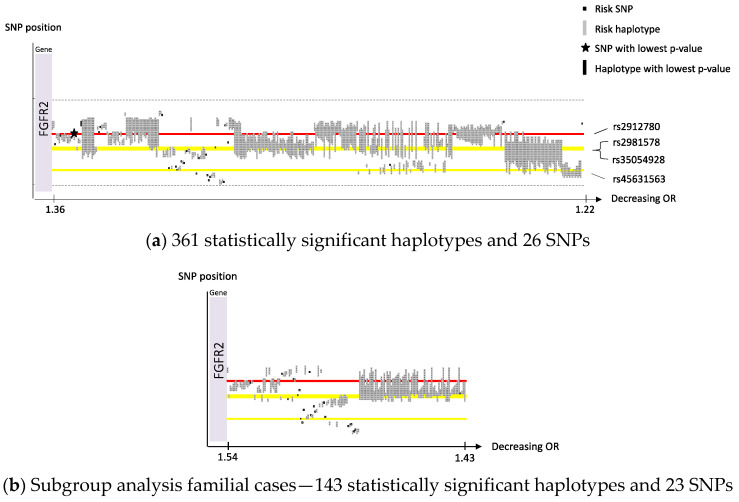
Chromosome 10, haplotypes and SNPs sorted on OR. (**a**) SNP position on the *y*-axis. Gene position for *FGFR2* is indicated along the *y*-axis. Corresponding OR on the *x*-axis. The red line indicates the position of the SNP with the lowest *p*-value. The three yellow lines indicate the position of three previous published BCAC SNPs within the area. The dashed lines indicate the position of the proximal and distal SNP of the haplotype of window 50 with the lowest *p*-value (non-significant); (**b**) axis, red and yellow lines illustrated as in (**a**).

**Figure 2 cancers-14-01206-f002:**
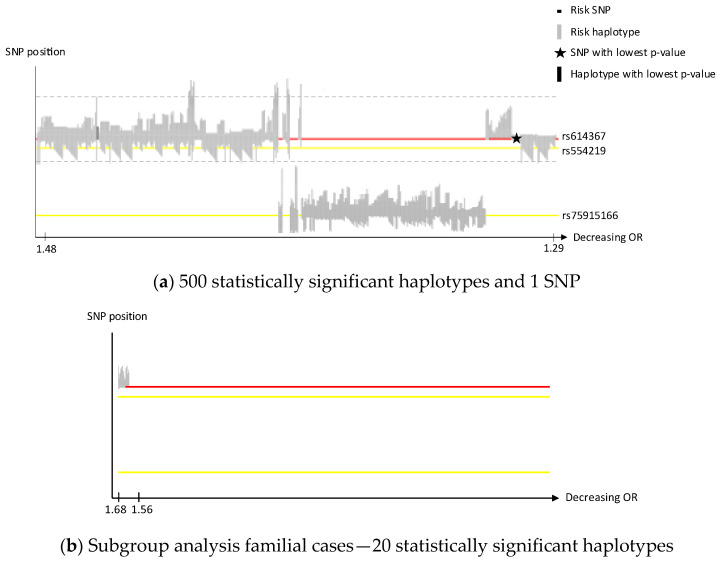
Chromosome 11, haplotypes and SNPs sorted on OR. (**a**) SNP position on the *y*-axis. Corresponding OR on the *x*-axis. No gene in the area. The red line indicates the position of the SNP with the lowest *p*-value. The two yellow lines indicate the position of two previous published BCAC SNPs within the area. The dashed lines indicate the position of the proximal and distal SNP of the significant haplotype of window 50 with the lowest *p*-value; (**b**) axis, red and yellow lines illustrated as in (**a**).

**Figure 3 cancers-14-01206-f003:**
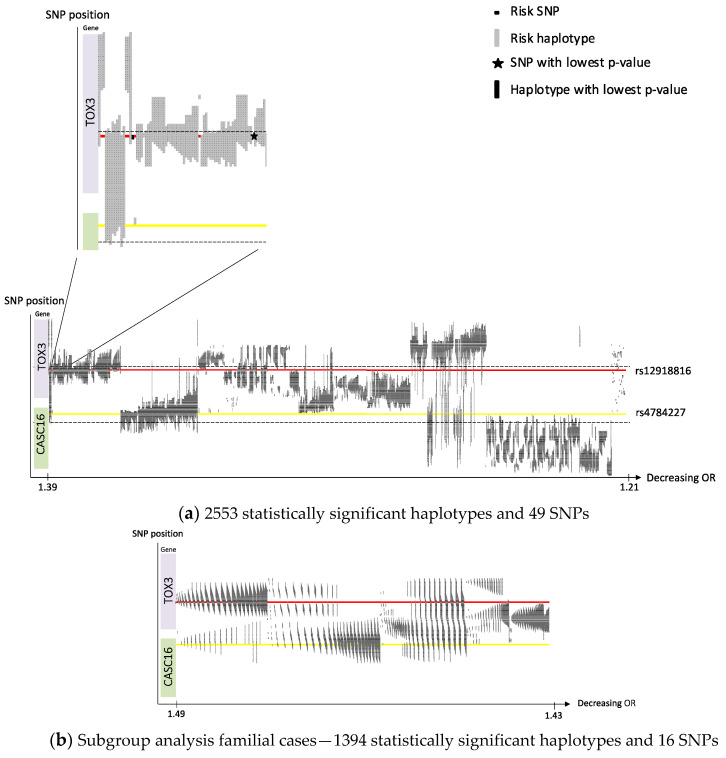
Chromosome 16, haplotypes and SNPs sorted on OR. (**a**) SNP position on the *y*-axis. Gene position for *FGFR2* is indicated along the *y*-axis. Corresponding OR on the *x*-axis. The red line indicates the position of the SNP with the lowest *p*-value. The yellow line indicates the position of one published BCAC SNP within the area. The dashed lines indicate the position of the proximal and distal SNP of the significant haplotype of window 50 with the lowest *p*-value; (**b**) axis, red and yellow lines illustrated as in (**a**).

**Table 1 cancers-14-01206-t001:** SNPs and haplotypes (window size 2–25 and 50) with lowest *p* value in four risk loci on Chromosome 8, 10, 11 and 16.

Locus	Genes	no. SNPs	SNP1-SNP2	F	OR (95%CI)	*p*-Value
8p21.2	*BNIP3L*	1	rs2034039	0.718	1.08 (1.00–1.15)	0.0299
		13	rs6995651-rs328097	0.0182	2.08 (1.60–2.70)	3.92 × 10^−8^
		50	-	-	-	-
10q26.13	*FGFR2*	1	rs2912780	0.412	1.36 (1.27–1.45)	1.67 × 10^−20^
		3	rs423753-rs10736303	0.421	1.36 (1.27–1.45)	8.76 × 10^−21^
		50	rs1047100-rs7899765	0.038	1.46 (1.24–1.71)	4.58 × 10^−6^
11q13.3	-	1	rs614367	0.161	1.31 (1.20–1.42)	5.45 × 10^−10^
		10	rs680618-rs614367	0.108	1.44 (1.30–1.59)	6.37 × 10^−13^
		50	rs7117818-rs673958	0.0812	1.45 (1.29–1.62)	1.81 × 10^−10^
16q12.1- 16q12.2	*TOX3* *CASC16*	1	rs12918816	0.259	1.37 (1.27–1.47)	3.26 × 10^−18^
		2	rs12918816-rs12929984	0.259	1.37 (1.27–1.46)	1.71 × 10^−18^
		50	rs11642645-rs1109951	0.214	1.38 (1.27–1.48)	5.20 × 10^−17^

Each genetic locus is presented with gene in the area (if any), number of SNPs included in haplotypes (no. SNPs), first (SNP1) and last SNP (SNP2) and corresponding frequency (F), Odds ratio (OR), and *p*-value for the significant SNP with the lowest *p*-value, the significant haplotype of window size 2–25 with the lowest *p*-value and the haplotype of window size 50 with the lowest *p*-value, respectively. Reference panel GRCH37 for SNPs.

**Table 2 cancers-14-01206-t002:** Published SNPs within the three loci on Chromosome 10, 11 and 16 in comparison with our data.

Locus	Genes	SNP	Allele	MAF		OR (95%CI)		*p*-Value		Ref.
				SWE	BCAC	SWE	BCAC	SWE	BCAC	
10q26.13	*FGFR2*	rs2981578	T/C	0.47	0.47	1.32 (1.23–1.40)	1.23 (1.21–1.25)	2.99 × 10^−17^	1.0 × 10^−114^	[30]
10q26.13	*FGFR2*	rs35054928	G/GC	-	0.4	-	1.27 (1.25–1.3)	-	3.5 × 10^−154^	[30]
10q26.13	*FGFR2*	rs45631563	A/T	-	0.05	-	0.81 (0.78–0.85)	-	9.1 × 10^−21^	[30]
11q13.3	*CCND1*	rs554219	C/G	0.14	0.13	1.25 (1.14–1.36)	1.21 (1.18–1.24)	6.50 × 10^−7^	5.8 × 10^−47^	[31]
11q13.3	*CCND1*	rs75915166	C/A	0.0733	0.06	1.38 (1.22–1.55)	1.28 (1.24–1.33)	1.36 × 10^−7^	4.1 × 10^−42^	[31]
16q12.1	*(TOX3)*	rs4784227	C/T	0.23	0.24	1.37 (1.27–1.47)	1.23 (1.2–1.25)	7.0 × 10^−88^	1.9 × 10^−30^	[32]

Abbreviations: SNP = Single Nucleotide Polymorphism; MAF = minor allele frequency; OR = Odds Ratio; Ref. = Reference; SWE = result from present Swedish GWAS; BCAC = Previous published Oncoarray results from BCAC Collaboration. Reference panel GRCH37 for SNP positions.

## Data Availability

Access to the data is controlled. Variants that fulfilled our selection criteria can be found in the Appendix A. However, Swedish laws and regulations prohibit the release of individual and personally identifying data. Therefore, the whole data cannot be made publicly available. The data that support the findings of this study are available from the corresponding authors upon a reasonable request.

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
