# Peer review of "A Swedish Genome-Wide Haplotype Association Analysis Identifies a Novel Breast Cancer Susceptibility Locus in 8p21.2 and Characterizes Three Loci on Chromosomes 10, 11 and 16"

_cancers, 2022, doi:10.3390/cancers14051206_

Round 1
Reviewer 1 Report
About ~50% of the breast cancer (BC) predisposition spectrum is still unexplained. In this manuscript, the authors use GWAS to search for another BC susceptibility loci in a large cohort of more than 3,000 Swedish BC patients and 5,000 controls. They identify a novel breast cancer risk haplotype at chromosome 8p21 and confirm three known loci at chromosomes 10, 11 and 16. In addition, the 8p21 haplotype is located within the BNIP3L gene that is a pro-apoptoptotic within the Bcl-2 family of proteins. This is the most interesting result of this study since, as indicated by the authors, the 8p21 region is considered to contain prostate cancer susceptibility genes and LOH has been reported in 20% of sporadic BC and 40% of familial BC. Therefore, a priori, BNIP3L is a good candidate as a BC susceptibility gene.
On the other hand, GWAS does not identify the BC causal variant within BNIP3L or other 8p21 genes, so my major concern is why this or other 8p21 genes has not been sequenced in this set of patients that shows association. Also, GWAS studies lack a functional correlation ,so transcriptomics data would provide critical data to establish the causality, and confirm or disregard BNIP3L as a BC susceptibility gene.
Also, the authors should indicate the implications of these findings in the clinical management of patients.
Minor comments
- The gene names should be in italics
- Line 332, typo “alles” → ”alleles”.
Author Response
Dear reviewer,
Thank you for your very thorough review of our paper and your valuable comments. Please see attached word-file with our responses to your valuable comments.
Yours sincerely,
Elin Barnekow, corresponding author

Reviewer 2 Report
This study intended to break through the limitation of SNP analysis to detect disease-specific haplotypes.
Authors need to check the following some minor points;
- Page 7, the last line: There is no Figure 5 in this manuscript. Is this a mistake of Figure 3?
- Page 8, line 11: Again, there is no Figure 6. Is this a mistake of Figure 3b?
- Page 8: The authors explain the sublocus 16q(a), (b), (c)…. However, it is a little bit difficult to find corresponding location in Figure 3. Please indicate these sublocus positions in the Figure 3a.
- Using the word “low risk loci” (on Chr10, 11 and 16) is a misleading and gives an impression of “preventive” function. Please consider this description.
Author Response

(The authors gave the same response as above.)

Reviewer 3 Report
Authors investigated breast cancer-related haplotypes and looked into SNP markers for genetic cancer susceptibility. A genome wide haplotype association study was conducted using sliding window analysis in 3,200 Swedish breast cancer cases and 5,021 48 controls. The study reports a novel breast cancer susceptibility locus in 8p21.1 (OR 2.08; p 49 3.92x10-8 ), confirmed 3 known loci in 10q26.13, 11q13.3, 16q12.1-2 and further identified novel sub- loci within these three loci.
The study is well designed and comprehensive.
There are only minor issues to address.
- Introduction section: lines 80-82: only general (vacuous) information is provided. Authors should extend this part and describe specific risk loci for Swedish breast cancers.
- Table 2: there is a problem with line numbering; line numbers got written over left side of the table.
- Discussion: the content of locus in 8p21.2 is well presented. However, three other loci were poorly presented. Authors can extend the description for them (putative target genes?). An additional figure can be shown here ( see here https://www.nature.com/articles/s41467-018-03411-9).
Author Response

(The authors gave the same response as above.)

Round 2
Reviewer 1 Report
No comments